# Defining upper extremity dominance: The contributions of hand preference and grip strength

**Mohamadreza Hatefi** [1]*, **Seyedeh Feriyal Mahdavi**[1], **Amirreza Abbasi**[2], **Farideh Babakhani**[1]

**1** Department of Sport Injuries and Corrective Exercises, Faculty of Physical Education and Sport Science, Allameh Tabataba'i University, Tehran, Iran, **2** Department of Sports Injury and Biomechanics, Faculty of Sport Sciences and Health, University of Tehran, Tehran, Iran

* hatefimohamadreza@yahoo.com

## Abstract

### Background

Upper extremity (UE) dominance is often defined by self-reported hand preference; however, this may not accurately reflect true functional or strength-based dominance. This study examined the relationship between writing hand, throwing hand, and maximal grip strength to assess how these measures align.

### Methods

Thirty-four healthy, recreationally active college-aged individuals reported their preferred writing and throwing hands and completed standardized grip strength testing. Associations among the variables were analyzed using Phi coefficients and chi-square tests.

### Results

A moderate, significant relationship was found between writing and throwing hand preference ($\varphi = 0.456$; $p = 0.008$), indicating general consistency across these subjective measures. However, no significant association emerged between grip strength dominance and either writing ($\varphi = 0.027$; $p = 0.876$) or throwing hand ($\varphi = 0.096$; $p = 0.574$).

### Conclusion

These results suggest that grip strength dominance may not correlate with commonly used indicators of hand preference, highlighting the need for task-specific definitions of dominance in clinical and athletic contexts. Consequently, employing such task-specific definitions allows for more accurate assessments and enhances the translational relevance of research findings in practical settings.

**Data availability statement:** The dataset underlying the findings of this study is included within the Supporting information files, and has been deposited in the Zenodo repository and is publicly available under a Creative Commons Attribution (CC BY 4.0) license. The data can be accessed at: https://doi.org/10.5281/zenodo.17436917.

**Funding:** The author(s) received no specific funding for this work.

**Competing interests:** The authors have declared that no competing interests exist.

## Introduction

Understanding upper extremity (UE) dominance is crucial in clinical, athletic, and rehabilitation contexts, as identifying the dominant hand enables clinicians and researchers to standardize measurements and compare variables both within and between subjects [1]. Therefore, it is vital to ensure that the dominant hand is accurately identified. However, while studies have indicated that the definition of the dominant hand can vary depending on the specific task [2–4] Coaches and researchers traditionally rely on self-reported preferences, such as the hand used for writing or throwing, to determine UE dominance. Although convenient, these measures are subjective and may not accurately reflect the true functional or strength-based dominance of the upper limb [2,5,6].

Hand grip strength is a widely accepted, objective measure of upper limb function and has been utilized to assess physical performance, detect asymmetries, and guide rehabilitation [7–9]. Despite its broad application, the relationship between hand preference (e.g., for writing or throwing) and grip strength remains unclear. In practice, it is often assumed that the dominant hand for writing or throwing will also demonstrate superior strength; however, this association has not yet been validated in the literature. Misclassifying limb dominance may limit the accuracy of assessments and interventions, particularly in sports biomechanics, neurological evaluations, and orthopedic rehabilitation. Furthermore, in complex movement tasks such as ball throwing—where coordination, explosive power, and instinctive limb selection are critical—individuals may exhibit hand preferences that differ from those identified through writing or grip strength assessments. Therefore, a more comprehensive approach is necessary to evaluate UE dominance by considering both subjective (hand preference) and objective (grip strength) indicators.

The relevance of this study lies in addressing a persistent gap in the definition and measurement of upper extremity dominance. Accurate identification of limb dominance is essential for designing individualized training programs, preventing injuries, and improving rehabilitation outcomes. Misjudging dominance can lead to biased data interpretation or inappropriate therapeutic strategies, particularly in athletic and clinical populations. Globally, research on upper-limb dominance has predominantly focused on Western populations, with limited data available from other regions, including Asia and the Middle East. Cultural, environmental, and training-related factors may influence hand use and functional asymmetries, underscoring the need for region-specific investigations. By examining physically active college students in our region, this study contributes valuable regional data to the broader global discourse on UP and functional asymmetry.

The purpose of this study was to investigate the relationship between commonly used indicators of UE dominance: preferred writing hand, preferred throwing hand, and maximal grip strength. This study aimed to determine whether subjective hand preference corresponds with objective measures of strength, thereby improving the understanding of dominance patterns in young, active adults. We hypothesized that a moderate to strong association would exist between writing and throwing hand preferences; however, grip strength dominance would not consistently align with either of the preference-based measures.

## Methods

### Participants

This study employed a cross-sectional design. A Phi-coefficient analysis was conducted using G*Power software version 3.1.0 (Franz Faul, University of Kiel, Germany), which calculated a minimum sample size of 34 participants to achieve a statistical power of 0.80, assuming a moderate effect size of 0.5 and a two-tailed significance level of 0.05. Accordingly, thirty-four healthy, recreationally active college-aged individuals were selected using purposive sampling, as they met the specific inclusion criteria relevant to the study objectives. All participants were undergraduate students in Physical Education and Sports Sciences, ensuring a baseline level of physical fitness. Recruitment was conducted over one month, beginning April 10, 2025, at the Faculty of Physical Education and Sports Sciences. To be eligible, participants had to be between 18 and 30 years old, either right- or left-handed, and engage in physical activity at least three times per week. All participants were healthy and physically active, and their enrollment in the Physical Education program further confirmed their regular participation in structured physical training and overall fitness. Individuals were excluded if they had a history of upper-extremity musculoskeletal injury or surgery within the past 12 months, any neurological disorder, or any other condition that could affect upper-limb performance or grip strength, such as chronic pain, arthritis, or tendon-related problems. The 12 months were selected to ensure complete functional recovery and to minimize any lingering effects that might influence muscle strength, coordination, or movement patterns during testing.

A study description was provided to all participants before the experiment, and all participants signed a written informed consent form. Each participant received a copy of the signed consent form for their records, and another copy was retained by the researcher in accordance with institutional ethical requirements. The study protocol received approval from the Ethical Committee of Allameh Tabataba'i University in Iran, and all procedures were conducted in accordance with relevant guidelines and regulations.

### Procedures

Participants reported to the sports biomechanics laboratory for a single testing session lasting approximately 45 minutes. Upon arrival, demographic data, including age, sex, height, and body mass, were recorded. Participants were then asked to identify their preferred writing hand. To assess grip strength dominance, isometric hand grip strength was measured using a calibrated digital hand dynamometer for both the right and left hands. During the test, participants were instructed to sit with their elbow flexed at 90 degrees, shoulder in a neutral position, and forearm in a mid-prone position, in accordance with the American Society of Hand Therapists' standard protocol [10]. Each participant performed three maximal grip contractions for each hand (maintained for 4–6 seconds), alternating sides with a one-minute rest interval between attempts to minimize fatigue. Notably, participants were verbally encouraged to exert maximal effort throughout the task. The highest value from each hand was recorded, and the hand with the greater value was classified as the stronger hand for that participant.

Following grip strength testing, participants were instructed to throw a 1 kg medicine ball repeatedly toward a target using one hand to assess their natural throwing preference. To minimize bias and elicit instinctive behavior, participants were not informed that their hand dominance was being evaluated during this task. In instances where participants utilized both hands, the examiner provided additional instructions, such as varying the throwing intensity, aiming accuracy, and changing the target location, to encourage the participant to rely on the hand they naturally preferred under these varied conditions.

All assessments were conducted by the same examiner to ensure procedural consistency. Before testing, participants engaged in a brief warm-up that included light upper extremity movements and practice trials with the dynamometer to familiarize themselves with the protocol.

## Statistical analysis

Descriptive statistics were employed to summarize participant characteristics and the frequencies of hand preference. Given the categorical nature of the variables, Phi coefficients were utilized to assess the strength of the association between preferred writing hand, preferred throwing hand, and hand grip strength. The strength of the association was interpreted based on the absolute value of the Phi coefficient ($\varphi$) as follows: 0.00–0.19 = very weak or negligible, 0.20–0.39 = weak, 0.40–0.59 = moderate, 0.60–0.79 = strong, and ≥0.80 = very strong. Chi-square tests of independence were also conducted to determine the statistical significance of the associations between variables. All analyses were performed using SPSS software Version 21.0 (IBM Corp., Armonk, NY), with the significance level set at $p < 0.05$.

## Results

Thirty-four healthy college-aged subjects (age = 25.5 ± 3.7 years; sex = 15 females, 19 males; mass = 73.05 ± 11.36 kg; height = 176.58 ± 8.94 cm) participated in this study. A series of Phi coefficient analyses were conducted to assess the relationships among preferred writing hand, preferred throwing hand, and hand grip strength. Twenty-eight participants wrote and threw with their right upper extremity; three wrote with their right hand and threw with their left; two wrote and threw with their left hand; and one participant wrote with their left hand and threw with their right upper extremity (see Table 1). The Phi coefficient ($\varphi = 0.456$; $p = 0.008$) indicated a moderate, statistically significant relationship between preferred writing and throwing hands, suggesting that most participants tend to use the same hand for both tasks.

In contrast, ten participants exhibited greater grip strength in their left hand, while twenty-four demonstrated greater strengths in their right hand. No significant associations were found between hand dominance and either writing hand ($\varphi = 0.027$; $p = 0.876$) (Table 1) or throwing hand ($\varphi = 0.096$; $p = 0.574$) (Table 1). These findings suggest that self-reported hand preference does not consistently correlate with hand grip strength dominance. Chi-square analyses corroborated these results, confirming that the observed and expected frequencies were not significantly different for either comparison involving grip strength ($p > 0.05$).

**Table 1. Two-by-two contingency tables displaying the frequency of preferred upper extremity for writing, throwing, and hand grip strength tasks (n = 34).**

**A. Writing Hand**

| Throwing Hand | | Left | Right | Total |
|---|---|---|---|---|
| | Left | 2 | 3 | 5 |
| | Right | 1 | 28 | 29 |
| | Total | 3 | 31 | 34 |

$\varphi = 0.456$; $p = 0.008$

**B. Writing Hand**

| Grip Strength | | Left | Right | Total |
|---|---|---|---|---|
| | Left | 1 | 2 | 3 |
| | Right | 9 | 22 | 31 |
| | Total | 10 | 24 | 34 |

$\varphi = 0.027$; $p = 0.876$

**C. Throwing Hand**

| Grip Strength | | Left | Right | Total |
|---|---|---|---|---|
| | Left | 2 | 3 | 5 |
| | Right | 8 | 21 | 29 |
| | Total | 10 | 24 | 34 |

$\varphi = 0.096$; $p = 0.574$

## Discussion

The present study investigated the correlation among commonly used indicators of UE dominance, specifically, writing hand, throwing hand, and grip strength in healthy recreationally active young adults. As hypothesized, we observed a moderate and statistically significant relationship between preferences for writing and throwing hands. However, grip strength dominance did not significantly correlate with either self-reported preference. These findings suggest that different measures of dominance may reflect distinct aspects of neuromuscular function, highlighting broader concerns regarding task-specific definitions of dominance across both upper and lower limbs.

Similar to the previous research who reported no significant relationship between preferred lower extremity for kicking and unilateral landing [11], our study found no correlation between hand preference and maximal grip strength performance. Carcia et al. emphasized the limitations of relying on a single, convenience-based definition of dominance (e.g., kicking leg) when it does not accurately represent the limb primarily involved in critical movements, such as landing movements that are associated with a higher risk of injury. Similarly, our findings suggest that preferences for writing or throwing may not adequately reflect the hand best suited for strength-based or functional capacity assessments. Notably, in our study, 10 of the 34 participants exhibited greater grip strength in their non-preferred hand, regardless of whether they were writing or throwing. This discrepancy aligns with previous research indicating that hand preference is not a reliable predictor of grip strength. For instance, it is reported a significant overall correlation between hand preference and grip strength [12]; however, they also reported that 24% of individuals demonstrated greater strength in their non-preferred hand. Moynihan et al. [13] observed no significant correlation between hand preference and strength, although they did find that hand preference was associated with skill-based tasks among right-handed males. Provins et al. [14] similarly noted that differences in hand performance were more strongly linked to specific task preferences than to general hand preference, with grip strength distributions showing complete overlap between left- and right-handers. These findings reinforce the idea that hand dominance is inherently task-specific and should not be assumed based solely on subjective reports. In light of these observations, the concept of dominance should be interpreted with greater nuance. Individuals may favor one hand for fine motor tasks (e.g., writing), while the opposite hand may demonstrate greater strength due to bilateral training or unconscious use in physical activities.

These insights have several important implications. In rehabilitation and performance settings, relying solely on hand preference to designate a "dominant" side may lead to misguided clinical decisions or training regimens, particularly when addressing strength deficits, bilateral asymmetries, or neuromuscular control. Instead, clinicians and researchers should carefully align the definition of dominance with the specific demands of the task or intervention. For example, when evaluating force production or muscular strength, grip strength testing provides a more direct and objective measure than self-reporting. The discrepancy between subjective and objective indicators may also reflect the complexity of neuromuscular adaptation. Individuals may rely on one limb for fine movement tasks (e.g., writing), while the other limb develops greater strength through non-specialized bilateral activities (e.g., sports, manual labor). These variations can further complicate the interpretation of limb dominance unless the selected metric aligns with the construct being evaluated.

We acknowledge that the current study has limitations that must be considered. As with the investigation conducted by Garcia et al., our study was limited by a homogeneous sample of healthy, college-aged individuals, which restricts the generalizability of our findings to other populations, such as older adults, children, or athletes engaged in high-demand sports. Furthermore, although our grip strength protocol adhered to standardized guidelines, it measured static isometric force rather than functional or dynamic upper limb performance, which could have provided a more comprehensive understanding of dominance. Additionally, the cross-sectional design limits causal inference and hinders the ability to track how dominance patterns evolve over time or with training.

## Conclusion

The current study reveals that, although writing and throwing hand preferences are moderately correlated, they do not correspond to grip strength dominance. These results suggest that grip strength dominance may not correlate with commonly

used indicators of hand preference, highlighting the need for task-specific definitions of dominance in clinical and athletic contexts. Consequently, employing such task-specific definitions allows for more accurate assessments and enhances the translational relevance of research findings in practical settings.

## Supporting information

**S1 File. Raw anonymized data underlying the findings described in this manuscript.**
(XLSX)

## Acknowledgments

The authors would like to express their gratitude to all participants and the staff of the Motion Analysis Laboratory for their collaboration in this study.

## Author contributions

**Conceptualization:** Mohamadreza Hatefi, Farideh Babakhani.

**Data curation:** Mohamadreza Hatefi, Seyedeh Feriyal Mahdavi, Farideh Babakhani.

**Formal analysis:** Mohamadreza Hatefi, Farideh Babakhani.

**Investigation:** Mohamadreza Hatefi, Seyedeh Feriyal Mahdavi, Amirreza Abbasi.

**Methodology:** Mohamadreza Hatefi, Seyedeh Feriyal Mahdavi, Amirreza Abbasi, Farideh Babakhani.

**Supervision:** Farideh Babakhani.

**Visualization:** Farideh Babakhani.

**Writing – original draft:** Mohamadreza Hatefi, Seyedeh Feriyal Mahdavi, Amirreza Abbasi.

**Writing – review & editing:** Mohamadreza Hatefi, Seyedeh Feriyal Mahdavi, Amirreza Abbasi.

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
