## [Decision Letter · Decision Letter 0]

7 Oct 2025

Dear Dr. Hatefi,

Thank you for submitting your manuscript to PLOS ONE. After careful consideration, we feel that it has merit but does not fully meet PLOS ONE’s publication criteria as it currently stands. Therefore, we invite you to submit a revised version of the manuscript that addresses the points raised during the review process.

We look forward to receiving your revised manuscript.

Kind regards,

Hasan Sozen

Academic Editor

PLOS ONE

Journal Requirements:

Reviewers' comments:

Reviewer's Responses to Questions

**Comments to the Author**

1. Is the manuscript technically sound, and do the data support the conclusions?

Reviewer #1: Yes

Reviewer #2: Yes

2. Has the statistical analysis been performed appropriately and rigorously?

Reviewer #1: Yes

Reviewer #2: Yes

3. Have the authors made all data underlying the findings in their manuscript fully available?

Reviewer #1: No

Reviewer #2: Yes

4. Is the manuscript presented in an intelligible fashion and written in standard English?

Reviewer #1: Yes

Reviewer #2: Yes

Reviewer #1: 3. Have the authors made all data underlying the findings in their manuscript fully available?

No

I perceive a restriction in the data availability from the statement of availability of data and materials. "The datasets utilized and/or analyzed during the current study are available from the corresponding

author upon reasonable request." The authors may need to clarify this.

Methods

The authors may have to specify if they used a sampling technique in this study. In this study, purposive sampling could be applicable and the reasons for its application stated clearly.

Ethics approval and statement of consent to participate

"All participants signed a written informed consent form." For such a study, it would be a good practice leave a copy of the signed consent form with the study participant, and one retained by the researcher. The authors may also clarify this.

Reviewer #2: I have re-attached the manuscript with track changes (where applicable) and comments.

The author is likely to have used AI to genenerate discussion, i have highlighted few areas,

Referencing style is heterogenrous. The Author shoule use one reference style in entire manuscript

**Do you want your identity to be public for this peer review?** For information about this choice, including consent withdrawal, please see our Privacy Policy

Reviewer #1: **Yes: ** Alyao Oscar Simon

Reviewer #2: **Yes: ** James Arinaitwe

---

## [Author Response · Author response to Decision Letter 1]

9 Oct 2025

Responses to the comments

We very much appreciated your encouraging and insightful comments. We have endeavored to respond to all suggestions and comments, which further improved the understanding and potential impact of our paper. We responded to the mentioned comments in both the “revised manuscript” file and this one. In the manuscript, responses to the first reviewer have been highlighted green and yellow for the second reviewer. Hope our effort meets the editorial board's expectations.

Sincerely Yours,

Authors.

Reviewer: 1

Authors: Dear Professor, thank you very much for your time and valuable comments. I truly appreciate your helpful feedback. I have done my best to revise the text based on your suggestions and believe the changes have strengthened the work.

Have the authors made all data underlying the findings in their manuscript fully available? No

Authors: We confirm that all raw data underlying the findings of this study have been uploaded as a supplementary file in the journal’s submission system.

I perceive a restriction in the data availability from the statement of availability of data and materials. "The datasets utilized and/or analyzed during the current study are available from the corresponding

author upon reasonable request." The authors may need to clarify this.

Authors: We appreciate the reviewer’s observation. To clarify, all raw datasets generated and analyzed during the current study have been “uploaded as a supplementary file” in the journal’s submission system and are therefore “fully available”. The previous statement regarding data availability has been revised accordingly to eliminate any ambiguity.

Methods

The authors may have to specify if they used a sampling technique in this study. In this study, purposive sampling could be applicable and the reasons for its application stated clearly.

Authors: We thank the reviewer for this helpful comment. The study indeed employed a purposive sampling technique, as participants were deliberately selected based on specific inclusion criteria relevant to the study objectives. The “Participants” section has been revised to reflect this clarification.

Ethics approval and statement of consent to participate

"All participants signed a written informed consent form." For such a study, it would be a good practice leave a copy of the signed consent form with the study participant, and one retained by the researcher. The authors may also clarify this.

Authors: Dear Professor, it is don and highlighted in the manuscript.

Reviewer: 2

Authors: Dear Professor, thank you very much for your time and valuable comments. I truly appreciate your helpful feedback. I have done my best to revise the text based on your suggestions and believe the changes have strengthened the work.

This is More of the Background of the study, rather than Introduction. Add relevancy of the study and or problem statement. Ad a paragraph on Global and regional picture.

Authors: Dear Professor, it is don and highlighted in the manuscript.

AI generated. Even when you use AI to generate a text, try to read through and use human language.

Authors: Dear Professor, thank you for your consideration. This section has been revised and highlighted in the manuscript.

Enrich this section.

Authors: Dear Professor, we have revised this section and highlighted the changes in the manuscript.

Why 6 months? a trauma for last 8 months may also affect performance of my hand.

Authors: We thank you for this valuable comment. Following your suggestion, the exclusion period has been extended to one year to ensure full recovery and minimize any residual effects on hand function. It should be noted that only one participant reported a previous hand injury and was therefore excluded from the study. This revision has been incorporated and highlighted in the manuscript.

How did you select participants? How physically were they fit? What measure was used to test or ascertain physical fitness.

Authors: Great comment. It is added and highlighted in the manuscript.

Repetitive citation. Reference 11 is Carcia, yet you cited Garcia. Secondly, you can either choose to use one referencing style, Either Havard, or APA, or Vancouva. Just be consistent.

Authors: It is done.

All is AI-generated.

Authors: Dear Professor, it is don and highlighted in the manuscript.

---

## [Decision Letter · Decision Letter 1]

24 Oct 2025

Dear Dr. Hatefi,

Thank you for submitting your manuscript to PLOS ONE. After careful consideration, we feel that it has merit but does not fully meet PLOS ONE’s publication criteria as it currently stands. Therefore, we invite you to submit a revised version of the manuscript that addresses the points raised during the review process.

https://journals.plos.org/plosone/s/submission-guidelines#loc-laboratory-protocols . Additionally, PLOS ONE offers an option for publishing peer-reviewed Lab Protocol articles, which describe protocols hosted on protocols.io. Read more information on sharing protocols at https://plos.org/protocols?utm_medium=editorial-email&utm_source=authorletters&utm_campaign=protocols .

We look forward to receiving your revised manuscript.

Kind regards,

Hasan Sozen

Academic Editor

PLOS ONE

Journal Requirements:

Reviewers' comments:

Reviewer's Responses to Questions

**Comments to the Author**

Reviewer #1: All comments have been addressed

Reviewer #2: All comments have been addressed

2. Is the manuscript technically sound, and do the data support the conclusions?

Reviewer #1: Yes

Reviewer #2: Yes

3. Has the statistical analysis been performed appropriately and rigorously?

Reviewer #1: Yes

Reviewer #2: Yes

4. Have the authors made all data underlying the findings in their manuscript fully available?

Reviewer #1: No

Reviewer #2: Yes

5. Is the manuscript presented in an intelligible fashion and written in standard English?

Reviewer #1: Yes

Reviewer #2: Yes

Reviewer #1: 4. Have the authors made all data underlying the findings in their manuscript fully available?

No.

Please state whether the data has been deposited in a public database / repository and indicate the data accessing license policies, which in this case should not be more restrictive than the Creative Commons Attribution (CC BY) license.

Reviewer #2: (No Response)

**Do you want your identity to be public for this peer review?** For information about this choice, including consent withdrawal, please see our Privacy Policy

Reviewer #1: **Yes: ** Alyao Oscar Simon

Reviewer #2: **Yes: ** James Arinaitwe

---

## [Author Response · Author response to Decision Letter 2]

24 Oct 2025

We thank the reviewer for this important comment. The full dataset underlying the findings of this study has now been deposited in the Zenodo repository and is publicly available under a Creative Commons Attribution (CC BY 4.0) license. The data can be accessed at the following DOI: https://doi.org/10.5281/zenodo.17436917

The Data Availability Statement in the revised manuscript has been updated accordingly.

---

## [Decision Letter · Decision Letter 2]

2 Nov 2025

Defining Upper Extremity Dominance: The Contributions of Hand Preference and Grip Strength

PONE-D-25-36856R2

Dear Dr. Hatefi,

We’re pleased to inform you that your manuscript has been judged scientifically suitable for publication and will be formally accepted for publication once it meets all outstanding technical requirements.

Kind regards,

Hasan Sozen

Academic Editor

PLOS ONE

Reviewers' comments:

Reviewer's Responses to Questions

**Comments to the Author**

Reviewer #1: All comments have been addressed

2. Is the manuscript technically sound, and do the data support the conclusions?

Reviewer #1: Yes

3. Has the statistical analysis been performed appropriately and rigorously?

Reviewer #1: Yes

4. Have the authors made all data underlying the findings in their manuscript fully available?

Reviewer #1: Yes

5. Is the manuscript presented in an intelligible fashion and written in standard English?

Reviewer #1: Yes

Reviewer #1: All comments have been fully addressed, and I believe the manuscript is ready for publication. I appreciate the authors for the time and resources they invested to ensure their work see merit. Thanks.

**Do you want your identity to be public for this peer review?** For information about this choice, including consent withdrawal, please see our Privacy Policy

Reviewer #1: **Yes: ** Alyao Oscar Simon

---

## [Editor Report · Acceptance letter]

PONE-D-25-36856R2

PLOS ONE

Dear Dr. Hatefi,

I'm pleased to inform you that your manuscript has been deemed suitable for publication in PLOS ONE. Congratulations! Your manuscript is now being handed over to our production team.

Kind regards,

on behalf of

Assoc. Prof. Hasan Sozen

Academic Editor

PLOS ONE